# Effect of Anti-Hypertensive Medication on Plasma Concentrations of Lysyl Oxidase: Evidence for Aldosterone-IL-6-Dependent Regulation of Lysyl Oxidase Blood Concentration

**DOI:** 10.3390/biomedicines10071748

**Published:** 2022-07-20

**Authors:** Rolf Schreckenberg, Oliver Dörr, Sabine Pankuweit, Bernhard Schieffer, Christian Troidl, Holger Nef, Christian W. Hamm, Susanne Rohrbach, Ling Li, Klaus-Dieter Schlüter

**Affiliations:** 1Institute of Physiology, JLU Giessen, 35390 Giessen, Germany; rolf.schreckenberg@physiologie.med.uni-giessen.de (R.S.); susanne.rohrbach@physiologie.med.uni-giessen.de (S.R.); ling.li@physiologie.med.uni-giessen.de (L.L.); 2Department of Cardiology, JLU Giessen, 35390 Giessen, Germany; oliver.doerr@innere.med.uni-giessen.de (O.D.); christian.troidl@innere.med.uni-giessen.de (C.T.); holger.nef@innere.med.uni-giessen.de (H.N.); christian.hamm@innere.med.uni-giessen.de (C.W.H.); 3Department Internal Medicine-Cardiology, Philipps-University of Marburg, 35043 Marburg, Germany; pankuwei@staff.uni-marburg.de (S.P.); schieffe@med.uni-marburg.de (B.S.); 4Department of Cardiology, Kerckhoff Heart and Thorax Center, 61231 Bad Nauheim, Germany

**Keywords:** hypertension, dilatative cardiomyopathy, endothelial cells

## Abstract

Lysyl oxidase (LOX) is a secretory protein that catalyzes elastin and collagen cross-linking. Lowering LOX expression and activity in endothelial cells is associated with a high risk of aneurysms and vascular malformation. Interleukin-6 (IL-6), elevated in hypertension, is known to suppress LOX expression. The influence of anti-hypertensive medication on the plasma LOX concentration is currently unknown. In a cohort of 34 patients diagnosed with resistant hypertension and treated with up to nine different drugs, blood concentration of LOX was analyzed to identify drugs that have an impact on plasma LOX concentration. Key findings were confirmed in a second independent patient cohort of 37 patients diagnosed with dilated cardiomyopathy. Blood concentrations of aldosterone and IL-6 were analyzed. In vitro, the effect of IL-6 on LOX expression was analyzed in endothelial cells. Patients receiving aldosterone antagonists had the highest plasma LOX concentration in both cohorts. This effect was independent of sex, age, blood pressure, body mass index, and co-medication. Blood aldosterone concentration correlates with plasma IL-6 concentration. In vitro, IL-6 decreased the expression of LOX in endothelial cells but not fibroblasts. Aldosterone was identified as a factor that affects blood concentration of LOX in an IL-6-dependent manner.

## 1. Introduction

Lysyl oxidase (LOX) is a secreted copper-containing amine oxidase that catalyzes elastin and collagen cross-linking and thereby helps to maintain the supporting architecture of the extracellular matrix in different organs and in the vasculature. Strong activation of LOX has been linked to cardiac dysfunction in animal models and is associated with diastolic heart failure and liver fibrosis in clinical settings [1,2,3]. Expression of LOX is also associated with increased aortic stiffness [4], whereas low LOX plasma concentrations or low extracellular activity of LOX are linked to aortic rupture and aortic aneurysm, specifically in the context of an activated renin–angiotensin–aldosterone system (RAAS) [5,6,7]. This suggests that anti-hypertensive therapies based on RAAS inhibition may normalize vascular LOX expression. 

LOX is strongly expressed in the endothelium of healthy subjects, but down-regulation is associated with endothelial dysfunction (reviewed by [8]). Pro-inflammatory cytokines such as tumor necrosis factor (TNF)-alpha and interleukin-6 (IL-6) repress the LOX expression in endothelial cells [9,10,11]. IL-6 is associated with blood pressure variation in hypertensive patients [12]. In experimental studies with mice, elevated IL-6 levels are correlated with a hypertensive response to psychosocial stress [13]. Thus, IL-6 is part of the pro-inflammatory phenotype that is associated with hypertension [14,15,16,17]. Under such conditions, the extracellular concentration of LOX likely declines, which favors a destabilization of the vessel wall. Therefore, normalization of LOX should be an aim of anti-hypertensive treatment, but the effects of anti-hypertensive medication on plasma LOX concentration are currently unknown. 

LOX activity and local concentration can be regulated on various levels, mainly by changes in local expression and proteolytic cleavage. N-acetyl-seryl-aspartyl-lysyl-proline (Ac-SDKP) is a naturally occurring endogenous bioactive tetrapeptide that is generated from thymosine β4 by prolyl-oligopeptidase and suppresses the angiotensin II-dependent up-regulation of LOX mRNA expression [18]. Whereas the full-length isoform of LOX can translocate into the nucleus and affect transcription, e.g., that of collagen III, secretion of the protein and extracellular cleavage leads to the formation of a smaller, enzymatically active extracellular oxidase [2]. As LOX is expressed in endothelial cells, its secretion should effect its plasma concentration. Cleavage of LOX can be activated by bone morphogenic protein-1 (BMP-1), tolloid-like-1 protein, and fibronectin [19]. 

To determine whether anti-hypertensive medication affects the plasma concentration of LOX, we assessed the LOX plasma concentration of hypertensive patients. We chose a cohort of patients with resistant hypertension whose blood pressure remained elevated despite intensive medical treatment with at least three different anti-hypertensive medications of different classes, including diuretics, at the maximally tolerated doses. This initial screening test was performed to detect potential candidates for involvement in the regulation of LOX secretion. As aldosterone antagonism showed the best correlation with LOX levels in this patient cohort, we subsequently verified the effect of aldosterone blockade on the plasma concentration of LOX in an independent second patient cohort. Here, patients with dilated cardiomyopathy (DCM) were chosen because a large proportion of them received aldosterone antagonists. Patients with known hypertension were excluded from this second cohort in order to examine whether the effect of aldosterone antagonism could be reproduced in a completely independent cohort. Finally, the observed relationship between aldosterone, IL-6, and LOX was confirmed by association studies of plasma concentrations and in vitro studies using isolated endothelial cells.

## 2. Materials and Methods

In this study, we analyzed blood samples from patients of the subproject 9a of the German Competence Network Heart Failure. Blood samples were taken at the time of inclusion. The investigation conforms to the principles outlined in the Declaration of Helsinki. All participants provided written informed consent. The study was approved by the ethic boards of the universities of Gießen and Marburg (patient cohort 1: approval code 199/55 (2 November 2015); patient cohort 2: approval code 115/04ff (6 September 2004)).

### 2.1. Study Design and Participants

This study was conducted to analyze the effect of anti-hypertensive medication on circulating concentrations of LOX. Two different prospective cohorts were used. In cohort 1, patients (n = 34) were initially enrolled for classification of resistant hypertension. Physical examination and laboratory tests were assessed for all patients. Patient data contain information about clinical history, age, sex, body mass index, blood pressure, and current medication. Office and 24 h ambulatory blood pressure measurements (ABPM) were performed in accordance with the current guidelines of the European Society of Cardiology for the management of arterial hypertension [20]. Patients with secondary origins of hypertension and those with known peripheral arterial disease were excluded. Patient cohort 2 included 37 subjects with DCM. Inclusion criteria were age between 18 and 70 years, left ventricular ejection fraction <45% as assessed by transthoracic echocardiography, and left ventricular end-diastolic dimension (LVEDD) ≥117% according to the formula of Henry. Exclusion criteria for cohort 2 were coronary stenosis >50% verified by coronary angiography, valvular heart disease, arterial hypertension accompanied by end-organ damage, or hypertension while on anti-hypertensive therapy. Patient data contain information about age, sex, and treatment with aldosterone antagonists. Blood samples were drawn from all patients at the time of inclusion and stored as plasma (cohort 1) or serum (cohort 2). All samples were processed immediately and frozen at −80 °C until assay. Patient characteristic of the two cohorts are given in Table 1. 

### 2.2. ELISA

The LOX concentration in plasma and serum samples was analyzed using a human LOX ELISA kit (E-EL-H0174; Elabscience, Houston, TX, USA) according to the manufacturer’s protocol. The limit of detection of the assay was 0.63 ng/mL. Aldosterone concentration in samples was analyzed using a human aldosterone ELISA kit (ab136933; Abcam; Cambridge, UK) according to the manufacturer’s protocol. The limit of the detection assay was 4.7 pg/mL. IL-6 concentration in samples was analyzed using a human IL-6 ELISA kit (D6050; R&D Systems, Minneapolis, MN, USA). The limit of the detection assay was 3 pg/mL. All measurements were carried out batch-wise on thawed samples by experienced staff blinded to patient characteristics.

### 2.3. Endothelial Cells and Fibroblasts

Male Wistar rats were used for the isolation of coronary endothelial cells. Cells were isolated as described before and grown for 2 days before use [21]. The purity of these cultures was >95% endothelial cells, determined by the uptake of acetylated low-density lipoprotein labeled with 1,1′-dioctadecyl-3,3,3′,3′-tetramethylindocarbocyanine perchlorate, contrasted with <5% cells that were positive for α-smooth muscle actin. Fibroblasts were also isolated from male Wistar rat hearts as described before [2]. These primary cultures were stained positively for vimentin and negative for the von Willebrand factor, indicating a purity of >95% fibroblasts. 

### 2.4. PCR

The mRNA levels of LOX were quantified by real-time RT-PCR using iQ-SYBR Green Supermix (Bio-Rad, Dreieich, Germany). Beta-2-Microglobulin (B2M) was used as a housekeeping gene to normalize samples. Total RNA was isolated from cells using peqGOLD TriFast (Peqlab, Biotechnologie GmbH, Erlangen, Germany) according to the manufacturer’s protocol. To remove genomic DNA contamination, isolated RNA samples were treated with 1 U DNAse/µg RNA (Invitrogen, Karlsruhe, Germany). Then, 1 µg of RNA was used to synthesize cDNA using Superscript RNase H Reverse Transcriptase and oligo (dTs) as primers. The sequences of primers used for determination had the following sequences: LOX: forward: ACA ACC GCA CTG CCT CTG CC; reverse: GCC TTG AGG CTC CAT CGC CG; B2M: forward: GCC GTC GTG CTT GCC ATT C; reverse: CTG AGG TGG GTG GAA CTG AGA C. Quantification was performed by the ΔΔt-method, as described before [22].

### 2.5. Statistics

The Shapiro–Wilk test was used to determine whether the data obtained in each group are normally distributed, and Levene’s test was used to assess whether data in two groups have a comparable variance. Group comparisons were made with either two-sided *t*-tests or the Welch test, as appropriate. Exact *p*-values are given, and differences having *p*-values < 0.05 are considered to be significant. Association studies were performed by linear regression analysis (Pearson correlation). 

## 3. Results

### 3.1. Drug Screening in Cohort 

The plasma concentration of LOX was quantified in 34 samples from patients diagnosed with resistant hypertension (Table 1). These patients (cohort 1) were currently on medication with between one and nine different anti-hypertensive drugs. Table 2 summarizes the mean LOX concentrations for the different treatment regimens. The data are not corrected for differences in sex, age, body mass index, blood pressure, or co-medication. We tested the null hypothesis that drugs have no effect on the LOX concentration. The null hypothesis was rejected with a *p*-value of < 0.05. With this definition, the null hypothesis was rejected only for patients on aldosterone antagonists. In these patients, plasma concentrations of LOX were increased with aldosterone blockade (*p* = 0.027).

### 3.2. Effect of ACE/AT1 Blockade, ASA, and Statins

From the screening (described in Table 2), it remained unclear whether ACE inhibition/AT1 blockade, statins, or ASS affect LOX blood concentrations. Although the differences were not significant based on the pre-defined criteria of *p* < 0.05, the effect sizes were large (ACE/AT1), or at least medium-sized (Statins, ASS). Therefore, we analyzed the data again but removed patients receiving aldosterone blockade (Figure 1). Furthermore, we tested the hypothesis that sex affects the LOX concentration in response to these three drugs more specifically because recently, sex-polymorphism in vascular expression of LOX has been noted [23]. The data suggest that the effect seen in patients with ACE/AT1 treatment or ASA treatment is mainly experienced by male patients, whereas no sex-dependent effect was seen for statins. Nevertheless, as even in male patients not receiving aldosterone inhibition, the effect of ACE/AT1 blockade or ASA remained below the pre-defined level of significance, allowing us to conclude that aldosterone is the main contribution to the changes in blood concentration of LOX.

### 3.3. Validation of Aldosterone Antagonism Effect in Cohort 2

The effect of aldosterone antagonism on the plasma concentration of LOX was re-evaluated in a second, independent patient cohort. We used this second cohort for two reasons. First, in cohort 1, only 5 of 34 patients were on aldosterone antagonists. Second, we wanted to verify the initial finding in a completely independent cohort. The patients in this cohort were diagnosed with DCM. Patients’ characteristics are given in Table 1. As already shown for cohort 1, patients treated with aldosterone antagonists had higher serum LOX concentrations (Figure 2).

### 3.4. Participation of IL-6 in Reduction of LOX Levels

The aforementioned data suggest that aldosterone down-regulates the vascular expression of LOX, leading to reduced blood concentrations. As mentioned in the introduction, IL-6 may be a candidate that triggers this effect in various tissues. To analyze whether IL-6 reduces the transcript expression of LOX in endothelial cells, isolated rat endothelial cells were cultured and stimulated with IL-6 (10 ng/mL) for 24 h. As outlined in Figure 3, IL-6 decreased the expression of LOX in endothelial cells with a Cohen’s d effect size of 1.86 (0.59–3.09, 95% confidence interval). In contrast, IL-6 did not suppress IL-6 mRNA expression in cardiac fibroblasts (Cohen’s d effect size: −0.019 (−1.32–0.95)). 

Our data suggest that aldosterone induces the release of IL-6 that then down-regulates vascular expression of LOX, leading to lower blood concentration of LOX. Finally, we proved this suggestion by correlating the blood concentrations of aldosterone with that of IL-6 in the patients of cohort 2 without aldosterone inhibition (Figure 4). Both concentrations correlated to each other (r = 0.608; *p* = 0.021, n = 14 patients). 

## 4. Discussion

The main finding of this study is that aldosterone blockade is associated with normalized blood LOX concentration. This finding was validated in two totally independent patient cohorts diagnosed with hypertension or DCM. This suggests that aldosterone down-regulates vascular expression of LOX. There is no evidence in the literature that aldosterone directly modifies the expression of LOX. However, aldosterone can activate the NFκB pathway [24] that is required to induce expression of IL-6. There is also direct evidence that aldosterone induces IL-6 in endothelial cells [25]. Upstream of NFκB, activation of TRAF3IP2 is involved in the signaling cascade that links aldosterone to induction of IL-6 [1,26]. The pro-inflammatory cytokine IL-6 suppresses the expression of LOX [9,10]. In our study, we show a positive correlation between blood concentrations of IL-6 and aldosterone and a direct effect of IL-6 on endothelial expression of LOX. Similar to pro-inflammatory cytokines, endothelial expression of LOX is suppressed by other stressors as well, such as low-density lipoproteins and homocysteine [27,28].

Paradoxically, cardiac induction of LOX may be stimulated by aldosterone via activation of the RhoA GTPase pathway in cardiac fibroblasts [29]. However, here we show that IL-6 represses LOX expression in endothelial cells but not fibroblasts, in accordance with prior findings that IL-6 does not repress LOX expression in fibroblasts [30]. Thus, aldosterone appears to increase the amount of LOX in cardiac tissue, with LOX subsequently contributing to the development of cardiac stiffness, whereas aldosterone triggers reduction in LOX expression in endothelial cells (probably via IL-6). Our data suggest that the blood concentration of LOX primarily reflects its vascular expression. Observed LOX elevation following treatment of patients with aldosterone antagonist agrees with the aldosterone-IL-6 positive correlation in blood of patients and with the inhibition of LOX expression in cultured endothelium in the presence of IL-6. Aldosterone blockade, however, should antagonize detrimental cardiovascular effects in both target organs. In the present study, aldosterone blood concentration was correlated with IL-6 concentration and aldosterone blockade was sufficient to normalize LOX concentrations.

IL-6 down-regulates the expression of LOX in other tissues as well. In lathyrism, a disease characterized by LOX inhibition, oxidative stress induces cytokines such as IL-6 [31]. Mechanistically, IL-6 is considered to induce direct alterations within the LOX promoter region [32]. Increased expression of IL-6 by homocysteine activated an IL-6-dependent cascade involving JAK2/Fli1 and the DNA (cytosine-5)-methyltransferase that down-regulates LOX by CpG methylation in the LOX promoter. High levels of homocysteine are commonly found in hypertensive patients with obesity, diabetes, and dyslipidemia [33]. In patient cohort 1 of our study, many patients received either statins or had a high body mass index. Therefore, although the concentration of homocysteine is not known in these patients, a similar correlation may occur in these patients as well. However, homocysteine may be a marker rather than an actor in the down-regulation of LOX, as it is affected by aldosterone blockade as well. In this study, we provide direct evidence that IL-6 down-regulates the expression of LOX in endothelial cells, consistent with our hypothesis that aldosterone decreases vascular LOX expression via IL-6. The latter one was supported by our finding that blood concentration of IL-6 and aldosterone showed significant correlation to each other.

Although our study identifies aldosterone as a key player in the regulation of plasma LOX concentration, it has some limitations. It is a cross-sectional study that is not normalized to the duration of aldosterone treatment, use of spironolactone or eplerenone, or dosage of the medication. Nevertheless, a common finding in two completely different patient cohorts is that the effect of aldosterone antagonism is completely independent of co-medication, sex, blood pressure, body mass index, or the additive effect of ACE/AT1 treatment. In addition, mechanistic insights from previous experimental studies suggest that aldosterone is an important regulator of vascular LOX expression [29].

We describe that ACE/AT1 blockade appears to elevate plasma LOX concentration (although failing to reach the pre-defined threshold value to reject the null hypothesis). However, the effect was less clear, restricted to men, and associated with more intensive co-medication. The different responsiveness of male and female patients was surprising, but taking into account that IL-6 is a key player in the vascular expression of LOX, this may be a relevant finding. Sex-dependent differences in IL-6 plasma concentrations have been reported previously [34]. In addition, ACE/AT1 blockade should also reduce the bioavailability of aldosterone. However, this patient cohort is a mixture of patients receiving AT1 blockers or ACE inhibitors. Recently, it was found that ACE cleaves Ac-SDKP [35]. Therefore, in patients taking ACE inhibitors, the Ac-SDKP plasma concentration is 4- to 5-fold higher, which should suppress LOX expression [35]. It may be that the effect of ACE/AT1 blockade on plasma LOX is balanced by the down-regulation of LOX mRNA expression via elevated Ac-SDKP. The beneficial effects of ACE inhibition on the release of aldosterone will lead to an increase in LOX expression in endothelial cells. Thus, ACE/AT1 antagonism may have mixed effects on the resulting LOX blood concentration. In any way, an elevation of blood concentration of LOX by aldosterone blockade is more sufficient than that of RAS inhibition. The most important finding of our study is that aldosterone blockade normalizes blood LOX.

Another potential candidate identified in our study that might affect blood LOX levels are statins (again, *p*-value did not allow us to reject the null hypothesis). Statins have been shown to directly affect vascular LOX expression in pigs, and the data in our present study are consistent with the idea that statins can also increase the plasma LOX concentration in men and women [36]. The mechanism for this response, which has been elucidated in pigs, is related to statin-mediated geranylgeranylated protein modification that leads to an activation of the RhoA/Rho kinase pathway and subsequently induces expression of LOX [36].

Limitations of this study come from the inherent sample collection of the two cohorts. Although a strength of our study is that the initially suggested modulator (aldosterone) was validated in a completely different patient collectives, there are, of course, differences in age, sample generation (plasma vs. serum), and storage time. It is therefore not surprising that the absolute levels differed between both cohorts. However, in the context of our study, it was more important to compare the individual samples within the cohorts.

Finally, ASA increased the blood concentration of LOX (again, the *p*-value does not allow us to reject the null hypothesis). In contrast to statins and ACE/AT1 blockade, there are no possible mechanisms related to this effect available in the literature.

None of the other treatment regimens, including beta-blockers, calcium antagonists, diuretics, or alpha-adrenergic antagonists, were associated with differences in the plasma concentration of LOX, indicating that the effect of aldosterone antagonism is rather specific. 

## 5. Conclusions

Our study provides evidence that aldosterone induces IL-6 that is responsible for a reduction in blood concentration of LOX. Furthermore, aldosterone antagonism is efficient to normalize LOX values. 

## Figures and Tables

**Figure 1 biomedicines-10-01748-f001:**
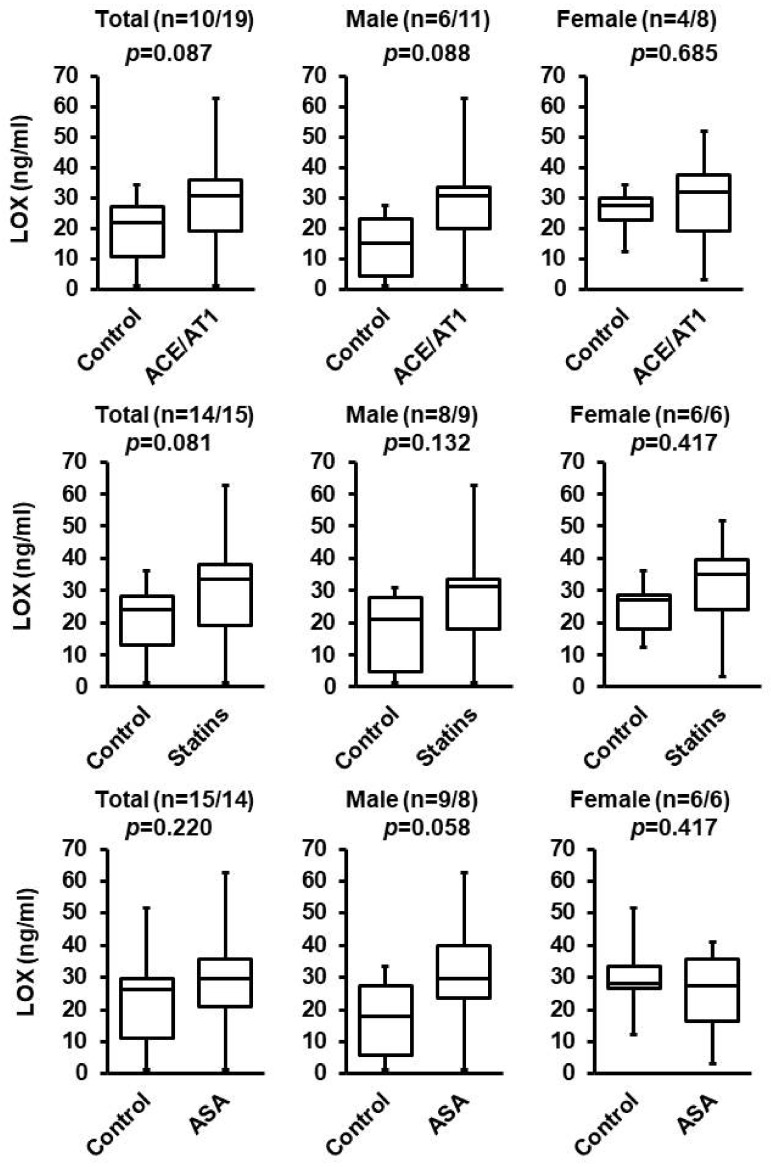
Effect of ACE/AT1 antagonism, statins, or ASA on plasma concentrations of LOX in hypertensive patients. LOX plasma concentrations in patients from cohort 1 (hypertension) either without (control) or with specified treatment (ACE/AT1 antagonism, statins, or ASA). Data are shown separately for the whole cohort, men only, and women only. The box and whiskers plots show the total range (whiskers), Q25, Q50, and Q75. Exact *p*-values are given and n-values are given.

**Figure 2 biomedicines-10-01748-f002:**
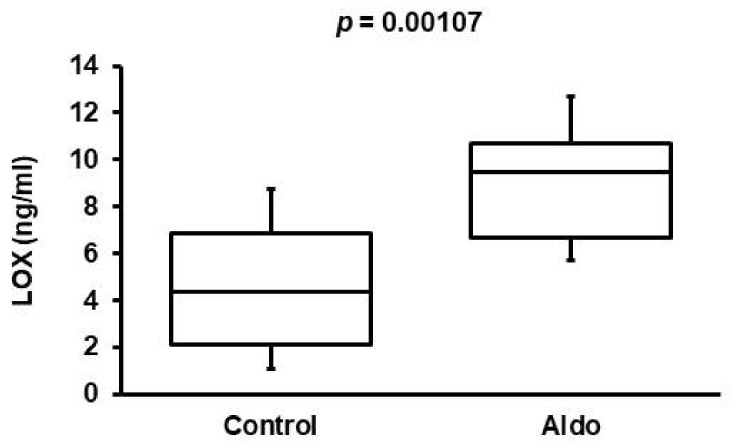
Effect of aldosterone blockade on serum concentrations of LOX in DCM patients. LOX serum concentrations in patients from cohort 2 (DCM) either without (control; n = 17) or with aldosterone blockade (Aldo; n = 19). The box and whiskers plots show the total range (whiskers), Q25, Q50, and Q75. Exact *p*-value is given.

**Figure 3 biomedicines-10-01748-f003:**
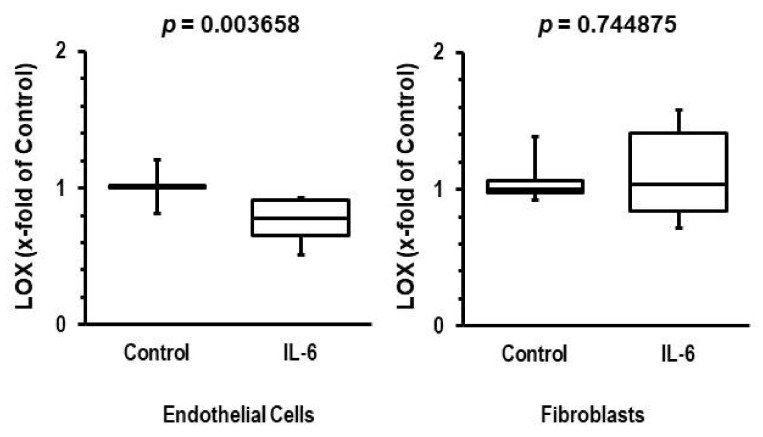
Effect of interleukin-6 (IL-6) on LOX mRNA expression in isolated microvascular rat endothelial cells and cardiac fibroblasts. mRNA expression of LOX was normalized to beta-2-microglobulin as a housekeeping gene. Mean expression of control cultures is set as 1. Cells were incubated for 24 h with IL-6 (10 ng/mL). The box and whiskers plots show the total range (whiskers), Q25, Q50, and Q75. Exact *p*-values are given (n = 6 cultures each).

**Figure 4 biomedicines-10-01748-f004:**
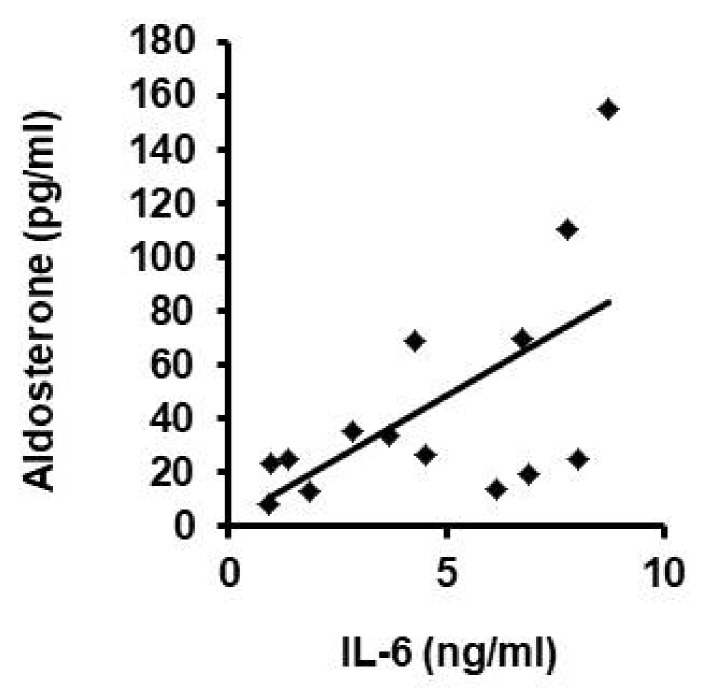
Correlation between blood concentration of IL-6 and aldosterone. IL-6 and aldosterone serum concentrations from patients of cohort 2 not receiving aldosterone blockade are plotted against each other. Linear regression analysis was performed by Pearson correlation (r = 0.608; *p* = 0.021).

**Table 1 biomedicines-10-01748-t001:** Patients’ characteristics.

	Cohort of Hypertensive Patients	Cohort of Patients with Dilative Cardiomyopathy (DCM)
Total number	34	37
Male/Female	20/14 (59%/41%)	20/17 (54%/46%)
BMI > 30	13 (38%)	
Diabetes	12 (35%)	
Current smoking	5 (15%)	
Age	67 ± 10 (years)	51 ± 13 (years)
BMI	29.8 ± 4.9 (kg/m^2^)	
P syst	153 ± 31 mmHg	
P diast	83 ± 19 mmHg	

**Table 2 biomedicines-10-01748-t002:** Effect of treatment on plasma concentrations of LOX in hypertensive patients (cohort 1).

Target	Control * (ng/mL)	Treatment (ng/mL)	Difference (ng/mL)	*p*-Value
Aldosterone antagonist	25.6 ± 15.8 (n = 29)	44.3 ± 18.5 (n = 5)	18.7	0.027
ACE inhibitor/AT1 blocker	20.3 ± 11.7 (n = 11)	32.2 ± 18.5 (n = 23)	11.9	0.067
Statins	22.6 ± 13.7 (n = 15)	32.9 ± 18.8 (n = 19)	10.3	0.094
ASA	23.6 ± 13.6 (n = 18)	33.7 ± 19.7 (n = 16)	10.1	0.098
Alpha-2-blocker	26.9 ± 17.6 (n = 23)	31.5 ± 16.8 (n = 11)	4.6	0.493
Diuretics	25.9 ± 20.2 (n = 9)	29.3 ± 16.3 (n = 25)	3.3	0.636
Renin inhibition	28.1 ± 18.8 (n = 29)	30.2 ± 5.3 (n = 5)	2.1	0.813
Beta-blocker	27.0 ± 19.6 (n = 10)	29.0 ± 16.5 (n = 24)	2	0.773
Calcium antagonists	28.4 ± 17.2 (n = 21)	28.3 ± 18.0 (n = 13)	−0.1	0.985
Alpha-1-blocker	28.6 ± 17.6 (n = 26)	27.7 ± 17.3 (n = 8)	−0.9	0.901

Values shown are mean ± standard deviation. * Refers to patients not taking the medication listed.

## Data Availability

Data can be provided from the corresponding author if reasonably requested.

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
