# Peer review of "Effect of Anti-Hypertensive Medication on Plasma Concentrations of Lysyl Oxidase: Evidence for Aldosterone-IL-6-Dependent Regulation of Lysyl Oxidase Blood Concentration"

_biomedicines, 2022, doi:10.3390/biomedicines10071748_

Round 1

Reviewer 1 Report

Effect of anti-hypertensive medication on plasma concentrations of lysyl oxidase: Evidence for aldosterone-IL-6-dependent 3 regulation of lysyl oxidase blood concentration

Schreckenberg et al., June 2022

Schreckenberg et al. assessed whether the LOX plasma concentration of hypertensive patients (with resistant hypertension) is affected by anti-hypertensive medication. Aldosterone antagonism was found to correlate with LOX plasma levels in the patients and subsequently verified in another cohort of patients (with DCM). They were solely chosen because of a broad aldosterone antagonists treatment within the cohort. In vitro studies were employed to confirm an association between LOX and IL-6. 

Schreckenberg et al.find that blocking aldosterone is associated with reduced blood LOC concentration, and that blood IL-6 concentration positively correlates with expression of LOX.

General

This is a rather small study. The authors present their data in a clear way and most information is available in the text or graphs. Some limitations to the study must be added to the text though and clarifications are needed in a few sections (see below). The authors could elaborate a bit more and more data could be presented. The authors nicely discuss their findings, but could, in my opinion, introduced more of their own data to this discussion. The samples obtained could be further analyzed for the levels of various modulators.

Materials and Methods

Line 108-110

Cohort 1 (resistant hypertension): samples = plasma 

Cohort 2 (dilated cardiomyopathy): samples = serum 

Samples are of different origin to between the two cohorts? What is the reason for this? 

Clotting factors in serum (or the platelets and cellular elements that contaminate plasma) could interfere with or alter your results. Please elaborate and discuss. Clinical control samples are difficult to obtain, but as both cohorts are based on patient data this should be added as limitations to the study. 

Line 110

How were the samples processed? Was any anticoagulant added to the plasma samples? If EDTA was used, please discuss the possibility that the calcium being chelated can inhibit enzymes/regulation thereof. 

Line 120

Have the authors thought about the decreased expression of LOX with increased age? 

From line 125

For the DCM patients is anything known about their blood pressure? It seems that much more information is given for Cohort 1.

Results

Line 180

The data shown are not corrected for the various parameters (age, sex etc). Line 208: I would suggest to either present the data differently or rephrase. It is not fully clear from the way the data are presented to the wordings and again to figure 1 what is actually found in the data pool. Line 208: “In patients treated with 208 ACE/AT1 blockade, LOX plasma concentrations were higher than those not taking this 209 medication; this was the case in men but not in women (Fig. 1).” This is also the case with other medications? And the graphical display distinguishes between the sexes (fig.1). Also, why did the authors decide to present the data between sexes? Why not BMI? Blood pressure? Other factors (hormonal) being of importance here?

Line 214-215

The authors claim that aldosterone blockade has a strong effect on plasma concentrations of LOX with only 5 out of 34 patients receiving aldosterone antagonists. This is a strong conclusion based on a very few n’s in a cohort. Please rephrase. 

Line 229

This was not shown for serum. The authors write that samples analysed from cohort 1 is plasma?

Figure 2

The levels of LOX in control/aldo-treated patients (cohort 2) is much lower than the values measured in cohort 1. Please elaborate. 

Also, why do the authors not present their data from men and women as with cohort 1? It would be interesting to see.

Line 238

As does the literature. Please cite.

Figure 4

Some points are unfortunately hidden by a label. Please redo.

Discussion

Line 268

‘Normalized’ or reduced…?

Line 271 - …

Ernesto Schiffrin already presented this idea in 2005; the effect of aldosterone on the vasculature. Please include this in the discussion. There might not be any direct evidence in the literature, but the idea of this effect is not novel as such. 

Minor

Line 167

Please add ‘s’ to statistic 

Line 170

Please correct to Levene’s test

Line 176

Please add ‘1’ to cohort to indicate which group are screened

Line 236

P-value, singularis

Line 250

Please correct to ‘cardiac fibroblast’

From the Discussion section the line spacing suddenly changes. Please keep it consistent. 

Reviewer 2 Report

1. n value for ASA treatment was only 5, and the significance was clear. Please provide the statistical analysis procedure.

2. What's the purpose put calcium antagonists and alpha 1-blocker in the target? The difference was negative. What did  they mean?

3. Direct evidence for alderserone on IL-6 and LOX was required.

Round 2

Reviewer 1 Report

I have no further comments. The authors have adressed the concerns raised. 

Author Response

We thank the reviewer again for evaluating our mansucript.